# Co-Hydrothermal Carbonization of Grass and Olive Stone as a Means to Lower Water Input to HTC

Rocío García-Morato [1], Silvia Román [1,*], Beatriz Ledesma [1] and Charles Coronella [2]

1. Applied Physics Department, University of Extremadura, Avd. Elvas, s/n, 06006 Badajoz, Spain; gmgrocio@unex.es (R.G.-M.); beatrizlc@unex.es (B.L.)
2. Chemical and Mineral Engineering, University of Nevada, 1664 N Virginia St, Reno, NV 89557, USA; coronella@unr.edu
* Correspondence: sroman@unex.es

**Abstract:** One drawback of biomass hydrothermal treatment (HTC) is the need of a water supply, which is especially important in the case of lignocellulosic biomass. This study has investigated the synergy resulting from co-HTC of two residual biomass materials that significantly differ in their physico-chemical compositions: (a) olive stone, OS, a hard and high-quality biomass, with low N content, whose potential to give a high heating value briquette by HTC has been proven, and (b) fresh grass pruning, GP, as it is gathered from gardens, with a high water content, moderate N fraction, and low calorific value. The work specifically focuses on the water saving that can be attained when the liquid product produced by one of them (grass, with 80% of moisture) can supply part of the water needed by the other (olive stone) when both are subjected to HTC simultaneously. It was found that, when instead of water, an additional amount of fresh GP is added (in particular 40 out of 110 g of water was provided by 54 g of GP), and a more basic processing water is obtained (pH of co-HTC increased by 40%, in relation of single OS processes). This in turn did not have a remarkable effect on OS final SY at any of the two temperatures studied (200 and 220 °C), not on the C densification. Other features such as N content of resulting OS hydrochars showed a rise in the case of hybrid processes, from 0.2% to 3.3%. Other features that were affected on OS HTC products because of the presence of the GP in co-HTC were the HC surface structure, hydrophobicity, and the presence of surface functionalities and their thermal stability towards pyrolysis; processing water also showed changes on mineral content when both biomasses there blended. Proving that a biomass like OS can be hydrothermally treated by a hybrid process involving less water, without being detrimental in terms of final SY and energy densification, can open a field of research aimed to make HTC processes more efficient in terms of hydric balance.

**Keywords:** co-hydrothermal carbonization; water saving; fresh biomass processing

## 1. Introduction

With the awareness of the need for sustainable use of natural resources, hydrothermal treatment (HTT) of biomass waste for energy and resource recovery has received increasing attention. In this context, hydrochars (HCs) produced by hydrothermal carbonization (HTC) of biomass have proven their effectiveness as fuels, adsorbent precursors, catalysts, or fertilizers.

The water supply could be one of the factors hampering the large-scale commercialization of HTC, given the global water crisis that is expected to worsen in the next decades, involving water shortages in an increasing number of many countries. Although the biomass reactant contains water (depending on the feedstock, in a wide range between 15–90%, $w/w$), typically, this is not sufficient for HTC, and a variable amount of water is needed.

On the other hand, the PW obtained by HTC is characterized by a high chemical oxygen demand (COD) due to the presence of different compounds. Although some of

these compounds are highly valuable chemicals (such as 5-hydroxymethylfurfural, levulinic acid, furfural, and products containing nutrients such as phosphorus [1], some others may have potential toxicity as phenols, furans, or N-heterocyclic compounds. The presence of these compounds needs proper management and are seen today as one important downside of a full-scale operation of HTC. Several successful alternatives have been proposed so far [2]; however, rather than using costly processes, saving the water volume needed to properly degrade feedstock is the most appealing option.

A reduction in the amount of water required can be achieved by recirculating HTC PW produced during HTC. Although it depends on processing conditions, in general, PW recirculation is not detrimental regarding the heating value and solid yield (SY), while it can improve the energy efficiency in the process by promoting secondary HC formation [3].

Co-hydrothermal carbonization is a process of HTC in which two distinct biomass resources are reacted simultaneously. When the two reactants are selected carefully, the HC produced by co-HTC has synergistic traits resulting from each separate biomass.

A number of studies have used co-HTC to improve any specific property on one of the feedstocks; for example, different studies report that adding biomass can upgrade HCs from feedstocks that are associated with bad combustion behavior. In this way, some authors have used co-HTC to decrease the presence of heavy metals on swine manure HCs, adding corn stalk or sawdust to the reaction systems [4]. Sewage sludge HCs have also been improved by blending this waste with lignocellulosic biomass prior to HTC, finding an improvement not only in the combustion (heating value and ash content) properties, but also in the solid yield [5,6].

Other studies have investigated synergetic effects on both of the wastes fed into the reactor; that is to say, situations in which each precursor benefits from the other regarding any aspect of the HC or processing water (PW). In this way, some investigators have subjected PVC waste with lignocellulosic biomass to co-HTC to produce an HC with reduced chlorine content relative to the HC derived from the first and a lower mineral content relative to the second. In this frame, the works of Yao et al. [7] or Xiaoluan et al. [8], using bamboo or corncob, or the research of Xu et al. [9] that used cotton fibers, or the work of Wei et al. [10] with pomelo peel, can be highlighted. The enhanced dechlorination of PVC during co-HTC has also been reported using coal, which also improved its mineral content [11].

Other examples of the synergy resulting from co-HTC can be found. Blending iron sludge with fallen leaves has also proven to provide a positive effect for both parts, helping the formation of magnetite on the first material HC and improving the kinetics of the second [12–14].

The literature on co-HTC shows that the effect that blending has on the final properties of the reaction products depends on the composition of the biomass precursors used as well as on the mixing ratio [15]. The effect of HTC conditions (time and temperature), however, have scarcely been investigated.

To date, the focus of most published research on co-HTC focuses on the characteristics of the HC produced. Little information is available on the PW produced from co-HTC, nor on the effects of the quantity of water used in co-HTC. This is in contrast to work published on HTC of a single biomass. None of them, to the best of the authors' knowledge, mention the fate of water during the process, and all use previously dried biomass materials for HTC processing.

In this way, this work aims to investigate the dehydration potential of HTC on a high-moisture biomass (garden pruning residuals, GP) to supply, under different temperature conditions, part of the water needed for a separate low-moisture lignocellulosic biomass (olive stone, OS). The starting hypothesis to investigate is whether dehydration (and corresponding water production) of the GP consisting primarily of high-moisture cut grass can provide a similar energy densification to OS, and to explore if changes in water acidity and conductivity when blending the two biomasses will have an effect on their respective

solid yield and final surface properties of the HCs. For it, the following experimental design has been followed:

(a) Single HTC runs, using OS or GP, at 200 and 220 °C (20 h). For OS, water was added to the system while GP experiments were conducted without any added water supply and were intended to determine how much aqueous solution can be obtained as a result of the treatment under these conditions.

(b) Co-HTC runs, in which the amount of the water supplied was reduced, assuming that it could be replaced by the PW produced by simultaneous GP HTC.

Liquid and solid yields were determined and compared for the different scenarios, and HCs were therefore analyzed in terms of thermogravimetric degradation, elemental composition, higher heating value (HHV), surface functionalities, and surface morphology. The results obtained allowed for inferring information about possible modifications on the degradation processes and the existence of synergies were explored.

## 2. Materials and Methods

Fresh grass (GP) was cut (3–5 cm long) from urban gardens in the city of Badajoz (Spain) always about 1 h before the experiments during the month of October 2022. Every day the GP was used, its moisture content was determined according to the Technical Specification CEN/TS 1474-2 (norm) [16] to make sure its moisture content had not changed significantly because of environmental or uncontrolled aspects; the humidity content ranged from 79–82% ($w/w$).

Olive stone (OS) was supplied by a local cooperative and was used as provided, without drying, cleaning, or grinding. Its average grain size was 2–5 mm and its humidity percentage was 10.4%.

### 2.1. HTC Processes

HTC processes were performed in a stainless-steel autoclave (Berghof, Berlin, Germany) provided with a 0.2 Teflon vessel (unstirred). Figure 1 shows a scheme of the experimental design. For single HTC processes, only one biomass was used; in the case of OS, a mass of 5 g and 110 g of tap water at room temperature were added to the reactor. For GP, no water was used and only fresh biomass (54 g), just after being cut from the gardens, was introduced in the system. In the case of hybrid HTC processes (co-HTC), the two biomasses were added simultaneously to the reactor (5 g of OS and 54 g of GP), with an amount of water of 70 g.

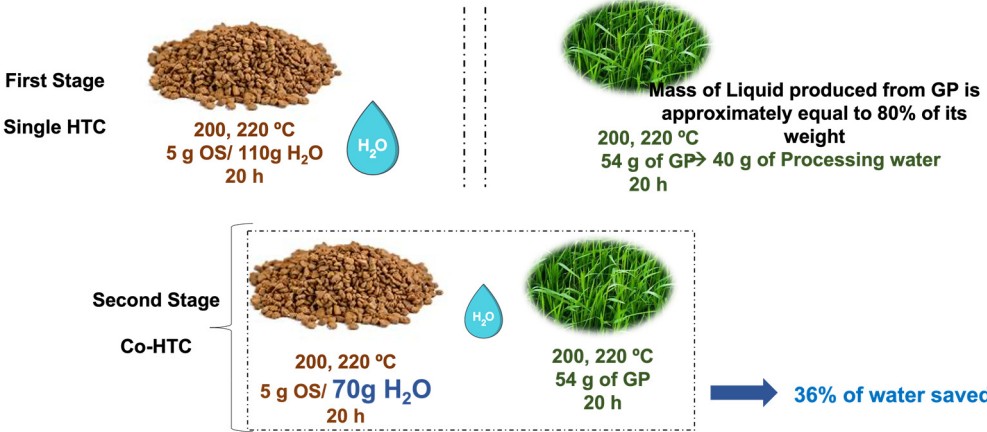

**Figure 1.** Single and Co-HTC reactions.

In all cases, the autoclave was then heated up in an electric furnace at 200 or 220 °C and kept in the furnace for 20 h. These reaction conditions were chosen based on previous HTC studies with other biomass sources using the same equipment [17].

When the reaction time was reached, the autoclave was removed from the furnace and subsequently placed in a cold water bath. Once room temperature was reached, the solid reaction products (one or two types of HCs, depending on the run) were filtered using Whatman filter paper (4 μm). After drying, the HC of each biomass was separately weighed and stored in flasks for further characterization. For co-HTC, the separation of two HCs was made manually; in this aspect, this work differs from others in that the properties of the HCs from each starting material are studied, not the whole solid product obtained (blend of the two HCs) after the process.

Solid yield (SY, %) for each material, was defined as the fraction of dried HC produced relative to the initial mass of dried biomass (OS or GP). In the case of GP, and additional SY was estimated, with the aim of providing a more realistic interpretation of the results, because this biomass was introduced into the reactor as it was cut (fresh); that is to say: without any humidity removal. This parameter was named SYF (Solid Yield taking the initial weight of fresh grass cut, without drying).

The pH and conductivity of the PW was measured (Crimsom) at the moment of collection. The amount of PW produced ($W_p$, in grams) in each run was calculated as the liquid collected after filtration plus the liquid lost during HC drying minus the amount of tap water added to the autoclave.

The following nomenclature was used: each single run was named HC_B_T, where B denotes the starting biomass (OS or GP) and T stands for temperature (200 and 220 °C). Processes in which the two biomass materials were used simultaneously were named co-HC_B_T, including the prefix co- and keeping the same meaning for B and T.

*2.2. HC Characterization*

The elemental composition (C, H, N, and S) of the two precursors was determined using a CHNS analyzer (LECO CHNS-932). O content was estimated as the difference between the sum of the analyzed elements to 100%; based on previous works, ash content was assumed to be negligible and not included in the equation. The HC fuel value was characterized by the HHV measured in a calorimetric bomb (PARR).

Thermal analyses (TGA and DTG) were performed using a thermobalance (STA 449 F3 Jupiter—Netzsch) coupled to a mass spectrophotometer (QMS 403D Aëolos III—Netzsch). A sample of around 10–12 mg was heated in the range 25–800 °C, under Ar atmosphere (100 mL/min) using a heating rate of 10 °C/min. Characteristic m/z relations were followed: $H_2O$ (18), $CO/N_2$ (28), and $CO_2$ (44).

The surface morphology of the samples was analyzed by scanning electron micrography (SEM, Hitachi, S-3600N, Krefeld, Germany) observation. The SEM samples were prepared by depositing about 50 mg of sample on an aluminum stud covered with conductive adhesive carbon tapes, and then coating with Rd–Pd for 1 min to prevent charging during observations. Imaging was performed in the high vacuum mode at an accelerating voltage of 20 kV, using secondary electrons.

Finally, the chemical functionality of the HC surfaces was evaluated by means of FTIR spectroscopy. FTIR spectra were recorded with a Perkin Elmer model Paragon 1000PC spectrophotometer (Waltham, MA, USA), using the KBr disc method, with a resolution of $4\,cm^{-1}$ and 100 scans (Perkin–Elmer 1720, Waltham, MA, USA). The composition of the crystalline part of the HCs was also analyzed by X-ray diffraction, using Bruker equipment (Warwick, RI, USA).

**3. Results**

*3.1. Solid Yield and Hydrochar Characterization*

The extent of HTC reaction was studied based on solid yield (SY, % wt/wt.) for each feedstock, and on the carbon densification of the HCs. As previously described, SY was calculated on a dry basis for both biomasses and in the case of GP, the parameter SYW, in

which the mass of fresh GP including its natural humidity just after it was cut, without drying, was used. That is to say, these two parameters were calculated as follows:

$$SY, \% = \frac{mass\ of\ dry\ HC}{mass\ of\ dry\ biomass} \cdot 100 \qquad (1)$$

$$WSY, \% = \frac{mass\ of\ dry\ HC}{mass\ of\ wet\ biomass} \cdot 100 \qquad (2)$$

A water balance allowed for determining how much water was produced as a result of the HTC reactions ($W_p$), and its acidity and conductivity properties gave insight about the process severity. These parameters have been included in Table 1.

**Table 1.** Solid yield (SY), elemental composition, HHV, (MJ/kg) of the HCs; amount of water produced $W_P$, pH, and conductivity. All data on solid products were made on a dry basis with the exception of WSY.

| | SY (%) | WSY (%) | $W_P$ (g) | C (%) | H (%) | N (%) | O * (%) | HHV MJ/kg | pH | cond. s/m$^2$ |
|---|---|---|---|---|---|---|---|---|---|---|
| **Raw OS** | -- | -- | -- | 44.8 | 6.2 | 0.1 | 49.1 | 17.2 | -- | |
| **Raw GP (fresh)** | -- | -- | -- | 8.2 | 10.0 | 0.9 | 80.9 | 2.9 | -- | |
| **Raw GP (dried)** | -- | -- | -- | 40.9 | 6.6 | 2.7 | 49.8 | 16.9 | -- | |
| **HC_OS_200** | 44.4 | | +0.8 | 64.5 | 5.0 | 0.2 | 30.3 | 25.8 | 3.1 | 445 |
| **HC_OS_220** | 40.4 | | −0.4 | 68.2 | 4.6 | 0.2 | 27.0 | 27.9 | 3.5 | 441 |
| **co-HC_OS_200** | 46.3 | -- | 40.1 | 61.6 | 5.63 | 3.3 | 29.4 | 25.9 | 4.5 | 366 |
| **co-HC_OS_220** | 41.3 | -- | 44.4 | 68.7 | 5,.5 | 1.3 | 24.6 | 30.2 | 4.8 | 356 |
| **HC_GP_200** | 68.5 | 13.7 | 35.0 | 58.7 | 5.34 | 4.0 | 32.0 | 22.1 | 4.8 | 345 |
| **HC_GP_220** | 51.5 | 10.3 | 39.8 | 62.2 | 5.4 | 4.0 | 28.34 | 27.3 | 4.4 | 331 |
| **co-HC_GP_200** | 69.6 | 13.9 | 40.1 | 59.6 | 5.3 | 3.4 | 31.7 | 25.9 | 4.5 | 366 |
| **co_HC_GP_220** | 60.6 | 12.1 | 44.4 | 63.6 | 5.5 | 3.7 | 28.3 | 27.9 | 4.8 | 356 |

* Balanced as: 100-C-H-N.

SY values of single HTC experiments show at first glance huge differences between the two precursors. OS presents an SY similar to that of other similar lignocellulosic precursors under the same reaction conditions: 40–45% [17,18], while GP as a result of its high moisture content (near 80%) had a weight reduction four times greater than the former material.

In reference to co-HTC runs, it can be seen from Table 1 that PW shows a modification on its acidity, as compared to any single experiment. A body of literature has looked at the effect of PW pH on HTC degradation processes of single biomasses, by adding acids or bases. For instance, other authors found that a change in the acidity (both a rise or a drop) had a significant effect on carbonization kinetics of pure cellulose but not on the carbonization extent [19]. Dissimilarly, other researchers found that an acid addition could enhance biomass degradation and result in a greater energy densification [20].

From our results, it seems that the slight increase that adding GP has on the PW in co-HTC (as compared to single OS HTC runs) has scarcely influenced the OS SY at both temperatures, although it might have affected the process kinetics. A different trend is found in the case of single GP HTC, for which GP SY is greater in the case of the hybrid process at 220 °C.

### 3.1.1. Elemental Analysis and High Heating Value

Independently of the process type (any single or Co-HTC process), a higher temperature enhances the feedstock degradation. This drop in SY, as in most of the works found in the bibliography for lignocellulosic biomasses, has a clear correspondence with the densification in C, mainly at the expense of the O removal.

The complexity of HTC processes applied to heterogenous biomass, making it very difficult to make general attributions to each reaction system. It is widely accepted that

hydrolysis is the starting point of the process, and that it should be enhanced with the increased ionic product of water, at greater temperatures [21]. Subsequent dehydration and decarboxylation reactions are catalyzed by H+ ions, and the wide variety of inorganic and organic acids can help further degradation processes. As this happens, water and reaction products also combine to form macromolecules that eventually can form nanoparticles that account to the solid recovery. In this way it is quite difficult, from the SY data, to infer whether a greater degradation of the parent material has taken place at a greater temperature, or if the lower SY is associated with a lower extent of condensation and recombination reactions. However, it is important to consider that the measured values correspond to the moment the autoclave is opened after cooling, and therefore degradation and recombination reactions beyond the set temperature and time might have happened.

It is clear, though, that degradation has been more efficient regarding C densification for the 220 °C than for 200 °C, a trend consistent with previous reports [17,18]. From the results presented in Table 1, it can be seen that using a higher temperature did not have a remarkable effect on the processing water pH, nor in the conductivity. However, it is important to consider that the measured values correspond to the moment the autoclave is opened, after cooling and therefore degradation and recombination reactions beyond the set temperature and time might have happened.

It is interesting to observe that HTC of both biomasses simultaneously by co-HTC produces two HCs and PW with properties distinct from those when the individual biomasses are reacted alone. In the case of liquid, an attenuation of PW acidity is found when adding GP to the OS system.

In relation to the HC elemental composition, a Van Krevelen diagram (Figure 2) can be very illustrative for delineation of reaction pathways. From the diagram, it can be seen firstly that temperature in the range studied involves a clear shift of the HCs towards the graph left bottom corner, that is significant for the hydrochars obtained from OS and very slight for GP, under any circumstance. Secondly, for OS co-HTC hinders dehydration (black points versus grey ones), providing higher H/C ratios. This effect is not found for GP, for which a slight decrease in O/C and increase in H/C is found when the other biomass is added. Any of these effects becomes more pronounced for the higher temperature.

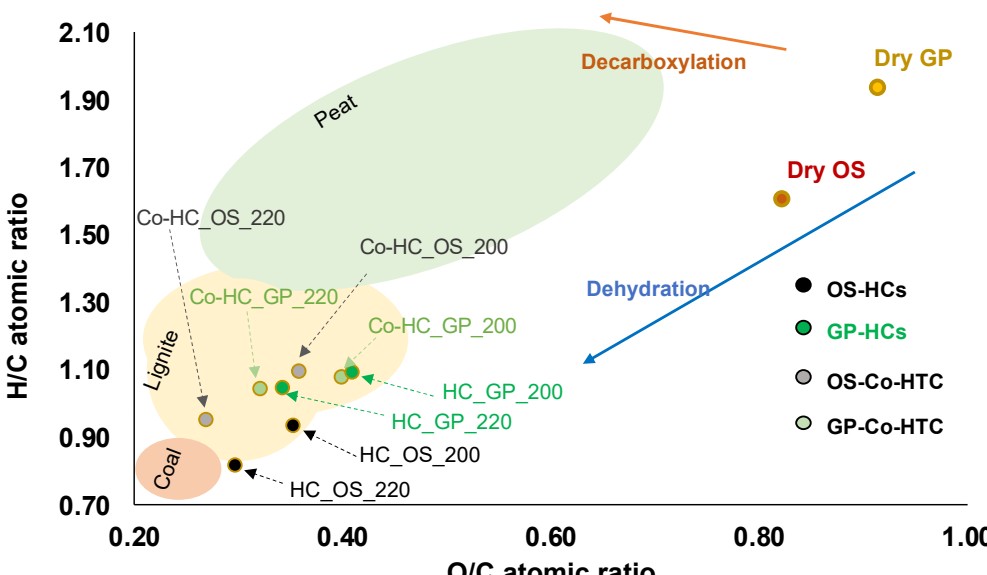

**Figure 2.** Van Krevelen diagram.

Lang et al. found that adding lignocellulosic biomass to swine manure promoted the dehydration reactions of the manure. In our case, using GP in conjunction with OS seems to slightly shift the process towards a greater prominence of decarboxylation [4].

The N content of a biomass, usually associated with its protein, determines the potential N capture of the HC, and this is clearly deduced from our results. GP, which has a significantly greater N proportion, yields HCs with N proportions of approximately 4%, while for OS, this value is close to 0.15%. Temperature had little influence on the N content of HCs produced in the range studied.

Using both materials at the same time (co-HTC) modifies the N concentration for both types of HCs, increasing that of OS and lowering that of GP. Many researchers have shown that deamination of protein can occur in HTC, resulting in ammonia present in the PW. In this case, the protein present in the GP undergoes deamination. In particular, if we follow the N capture value (that is, amount of N that remains on the HC in reference to the amount of N (g) previously found on the biomass, as calculated in reference [22], it can be seen that the adsorption of N compounds was specially enhanced in the case of OS; that is to say, attaining a greater condensation of N-compounds on the OS HC occurs when GP is added to the system.

The bibliography shows that in general, a rising temperature enhances protein decomposition, which generally results in a greater N content in the liquid phase. Separately, with time, some of the N-containing compounds at the liquid phase can be condensed or adsorbed on the HC. As a result of these two effects, the N capture on the HC can be more or less significant depending on the reaction conditions.

### 3.1.2. Thermogravimetric Analyses

Following the weight loss of biomass by thermogravimetry can be used to infer valuable information about its composition. In this study, several findings were deduced by comparing the TGA/DTG curves of the OS HCs obtained under plain (only water) and co-HTC conditions (Figure 3a).

Firstly, regarding simples HC_OS_200 and HC_OS_220, for which only OS was used in the system, the decomposition path is quite similar to the one reported in the literature for similar lignocellulosic materials [17]. Leaving behind the very slight weight loss associated with moisture (T < 150 °C), a sequence of mass loss trends of different slope is found in the TGA curves. Namely, a slow mass loss is then followed by a rapid one, and finally a slight mass decrease is still observed up to 850 °C.

It can be clearly observed that, although these three stages are present for all samples:

(a) the temperature ranges associated with each trend is shifted towards lower temperatures for the samples hydrocarbonized under milder conditions (200 °C), regardless of the run (single or co-HTC).

(b) co-HTC makes the HC less resistant to thermal mass loss than single HTC, at any of the studied temperatures, and also produces at each stage a greater mass loss.

This different behavior is indicative of the greater volatile content the OS HCs have when GP is added (Co-HC_OS as compared to HC_OS) to the system. This suggests that HTC of GP produces a greater concentration of organic molecules in the PW, which allows the deposition of a greater number of condensed macromolecules on the HC surface, which are then easier to decompose by pyrolysis in the TGA.

In general, lignocellulosic biomass subjected to TGA under inert conditions displays a more defined shoulder (that can even be an independent left little peak) in the central mass loss region, when the biomass has a moderate content in hemicellulose. In this way, cellulose degradation is generally ascribed to the central main peak, hemicellulose is associated with the definition of this left-side protuberance on the main peak, while lignin can have such an heterogenous composition that is usually associated with a very broad temperature range (150–750 °C) [23].

Our results suggest a lower proportion of hemicellulose and cellulose for HC_OS_220 (not only the peaks are located at a greater temperature for this sample, but also the peak intensity is lower) and also a more resistant lignin structure. After pyrolysis, both HCs attain a final weight loss of 0.46 and 0.54 for HC_OS_200 and HC_OS_220, respectively,

suggesting a larger amount of volatile matter in the former sample, as expected because of its softer former HTC degradation.

On the other hand, adding fresh grass to the HTC reactor has an effect on the HTC process, for the two temperatures applied (Co-HC200_OS and Co-HC220_OS). While degradation starts at the same temperature for analogous runs (i.e., made at the same HTC temperature) but exhibits a greater slope when the HC is made with GP (compare Co-HC200_OS to HC200_OS, and Co-HC220_OS to HC220_OS). The proportion of more easily degradable compounds is therefore higher for co-HTC samples, probably because of the deposition of additional secondary HC during these runs, as promoted by the greater condensable molecule concentration in these cases.

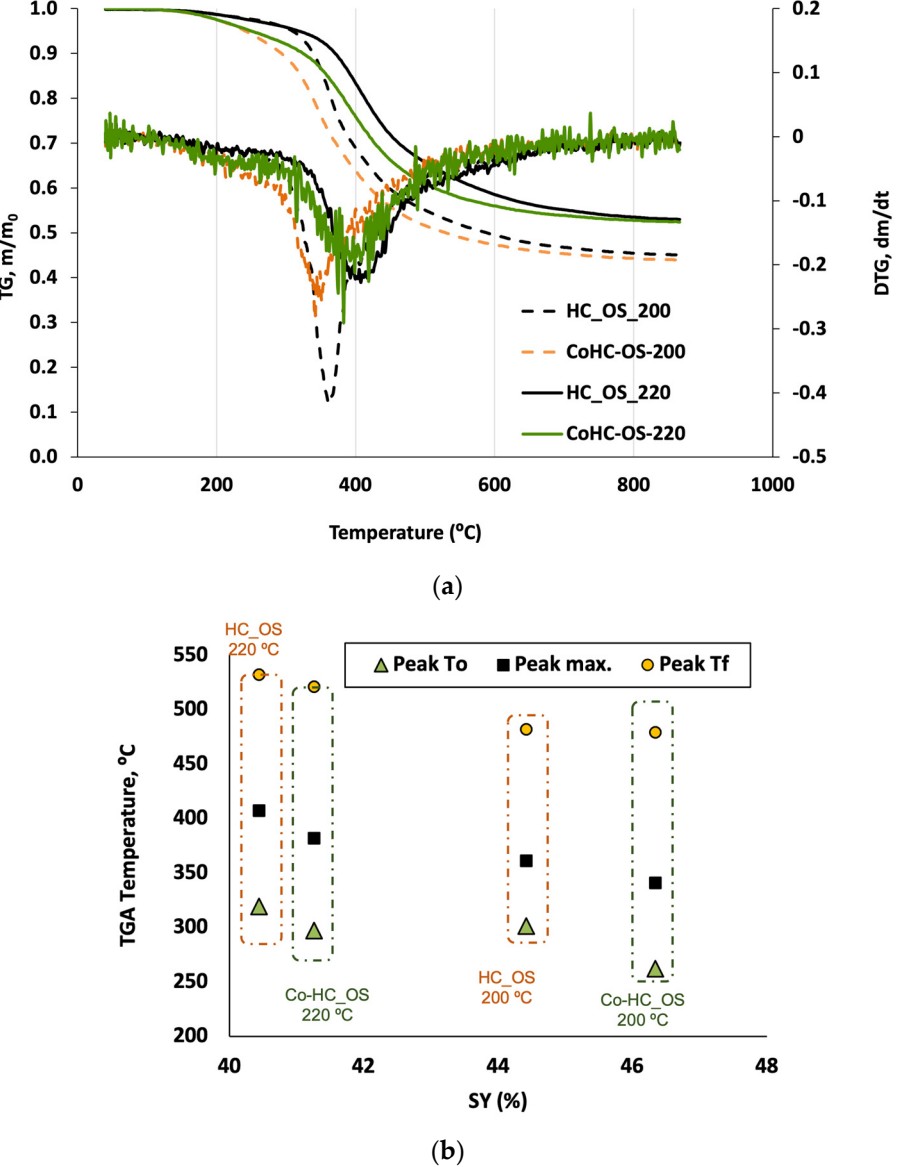

(**a**)

(**b**)

**Figure 3.** (**a**) TGA/DTG analysis of OS HCs from single and Co-HTC reactions and (**b**) characteristic TGA peak temperatures vs. HC solid yield.

It is very interesting, further, to follow how the SY of an HC has a relation to its thermal degradation. As shown in Figure 3b, the temperature of each characteristic peak, associated with the loss of volatile matter during pyrolysis follows the opposite trend of the SY (since a lower SY corresponds to a greater devolatilization; there is less matter susceptible to thermal degradation after further pyrolysis). This can be clearly observed from Figure 3b,

there one can see how the three temperatures that define the main degradation peak: (a) starting of the main peak, $T_0$; (b) its maximum value ($T_{max}$), and (c) the final peak T ($T_f$), for the four HCs prepared from OS. Any of these temperatures are displaced towards higher values for HCs that were prepared at 220 °C than at 200 °C.

Coupling mass spectrophotometry to TGA analyses allowed for inferring information about the composition of the gas evolved upon thermal degradation of the HCs. By the deconvolution of the emission curves, the amount of CO, $CO_2$, and $H_2O$ mg/mg, %) were estimated for the OS runs (both single and Co-HTC reactions) and are shown in Figure 4. In this Figure, we have plotted the number of gases released when HCs from GP, both in the case of single and hybrid HTC processes, are subjected to pyrolysis ($Y$ axis). In the $X$-axis, we have calculated a ratio that represents the amount of water recovered as a result of HTC (Wp, in g) in relation to the water that could be physically desorbed by drying that amount of feedstock (in g).

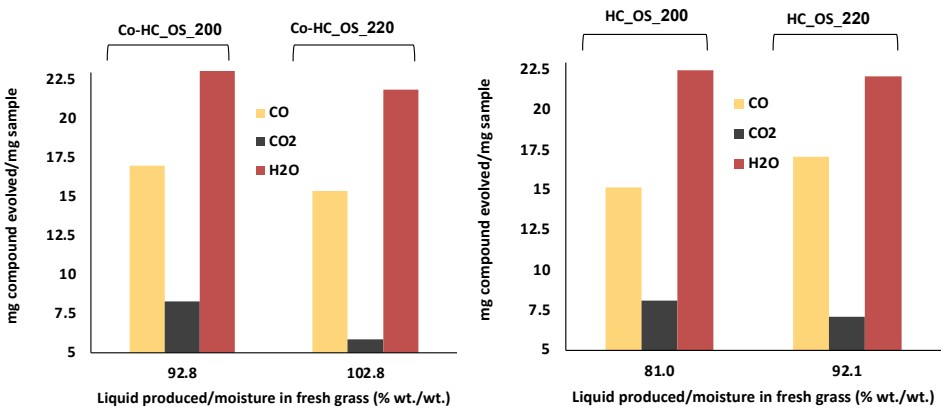

**Figure 4.** Percentage (%, $w/w$) of compounds released during pyrolysis towards the amount of liquid recovered in PW at 200 and 220 °C.

It can be seen that for those runs that involved a greater water production (220 °C), the amount of water that is released as a result of the corresponding HC pyrolysis is lower; both for single GP HTC and hybrid processes.

3.1.3. Surface Analyses

Surface morphology analyses provides complementary information supporting the hypothesis about the enhancement of a secondary HC formation for co-HTC runs, as well as the modification of its structure and chemical functionalities.

Uniform micrometer-sized spheres, clearly observed for HC_OS, were almost absent for OS HCs obtained by co-HTC (see Figures 5 and 6); that is to say, the presence of GP in the reaction medium either inhibited the LaMer nucleation reactions by which these spheres are formed [24], or either caused the destruction of the spheres after their formation.

In the case of OS, the size range of the spheres (with diameters in the range 2–18 μm) showed a relatively high degree of uniformity. Some of these spheres fuse at some points, in coherence with the results found for other organic materials once a given temperature is attained [24].

In reference to the co-HTC processes, SEM analyses showed a complete absence of microspheres. The surface of the OS HC displays the cellular original structure of the biomass precursor, together with the presence of a pointed coral-like second phase that is far from spherical.

Bearing in mind that this material was more vulnerable towards pyrolysis, the results suggest that the polymerization and condensation of PW fragments gave a more stable and ordered secondary HC when only OS was used on the system.

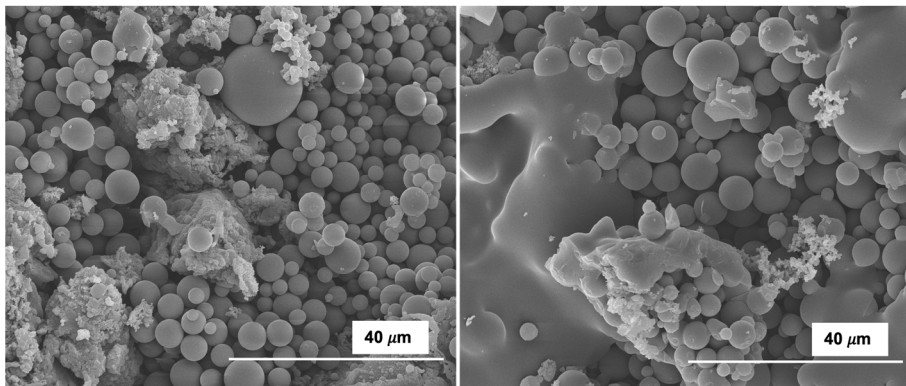

**Figure 5.** SEM micrographs for selected samples: HC200_OS (**left**); HC220_OS (**right**). Magnification: 3500×.

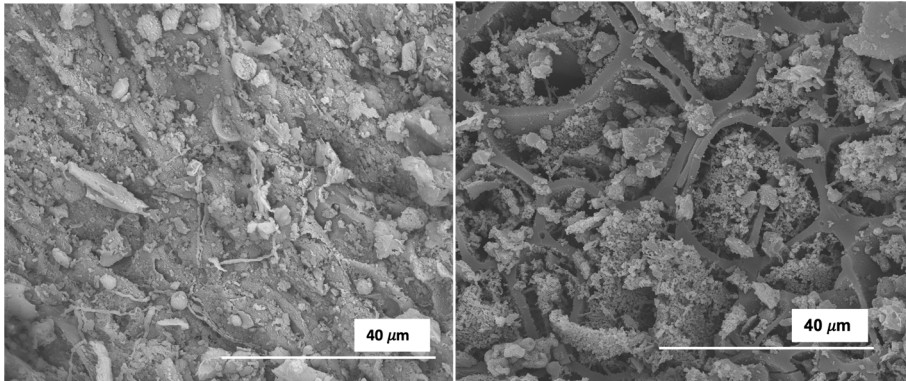

**Figure 6.** SEM micrographs for selected samples: Co-HC200_OS (**left**); Co-HC220_OS (**right**). Magnification: 3500×.

The potential changes on the surface chemical functionalities of OS HCs were evaluated by FT-IR analyses, spectra shown in Figure 7. As a whole, it seems that only slight changes are found when temperature is varied within the range studied, while the effect of incorporating GP to the system does not have a significant influence on the number or intensity of bands. The authors, in a previous work [3], found that recirculating PW from HTC of pine cones had an effect on surface chemistry only when temperatures higher than 200 °C were used; that is to say, a richer PW can influence the HC functional groups determined by this technique only to a limited extent. Other studies showed how the HCs from swine manure changed if lignocellulosic biomass was added to the liquid, finding an enhanced aromatization of the surface [4].

From our results, some specific comments can be made. For example, the wide band at around 3450 cm$^{-1}$, usually found in carbon materials, is associated with stretching vibration of (O-H) due to water adsorption, is more intense for HCs made at 200 °C (in consistence with lower dehydration), and does not show a difference when GP is added. The bands at around 2930 cm$^{-1}$ are slightly more intense in the case of Co-HTC samples (especially at CoHC-OS-220). In these two samples, a slight peak also appears in the left of these bands at about 2945 cm$^{-1}$.

Co-HTC samples have a band at 1060 cm$^{-1}$ that can be associated with characteristic C-O-C vibrations in the cellulose and hemicellulose, indicative that some holocellulose fragments and intermediate structures remained in the HCs, to a bigger extent than in single processes; this tendency has also been observed when using recirculating PW [3].

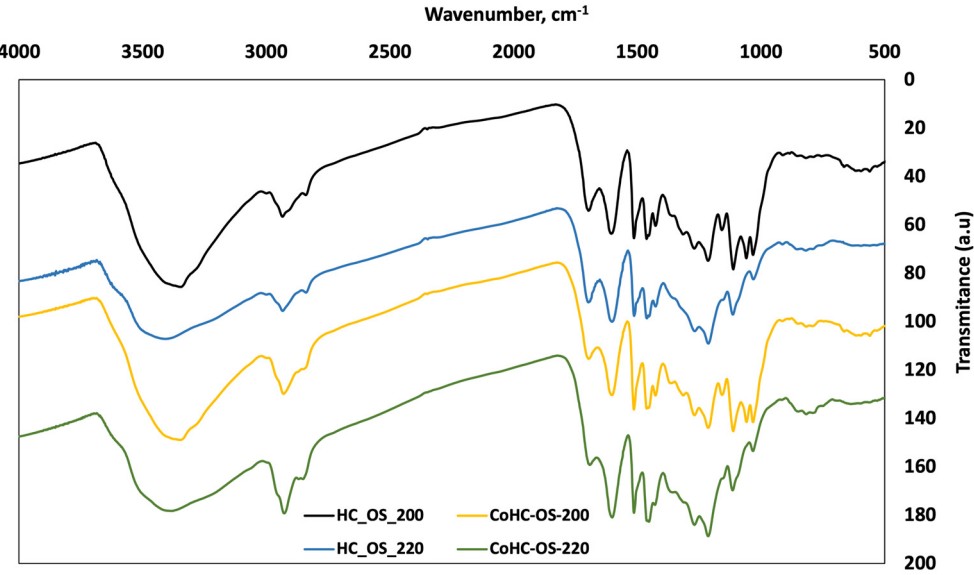

**Figure 7.** FT-IR spectra of OS HCs from single and Co-HTC runs.

*3.2. Process Water Production and Characterzacion*

Process water of both HTC and co-HTC experiments was analyzed in terms of metal concentration as well as elemental analysis (Table 2).

**Table 2.** Processing water characterization.

| | Concentration (mg/L) | | | | | | | Conc. (wt %) | |
|---|---|---|---|---|---|---|---|---|---|
| | **Mg** | **Al** | **P** | **Ca** | **Fe** | **NO$_2$** | **NO$_3$** | **C** | **N** |
| **HTC_OS_200** | 173.3 | 0.31 | 195.37 | 275.90 | 0.31 | 0.69 | 17.68 | 0.120 | 0.21 |
| **Co-HTC_OS_200** | 9.74 | 0.19 | 12.52 | 54.88 | 0.85 | <0.1 | 6.34 | 0.055 | 0.28 |
| **HTC_GP_200** | 488.62 | 0.81 | 605.73 | 62.33 | 0.61 | <0.1 | 100.36 | 0.22 | 0.43 |
| **HTC_GP_220** | 387.93 | 1.51 | 307.43 | 290.36 | 1.25 | <0.1 | 7.27 | 0.08 | 0.37 |

Starting with the effect of the temperature, it has been reported that PW acidity (generally higher as a result of enhanced hydrolysis) facilitates the deamination of aminoacids. This, in turn, should favour the production of ammonia and nitrate ions in the liquid phase. However, if reaction time is high, as in the case studied here (20 h), the recombination of molecules to yield macromolecules and eventually secondary HC as well as the adsorption of N-containing compounds on the HC surface is very likely to occur, thus giving rise to a decrease in the measured concentration of these compounds on the processing water.

The results shown here for single HTC runs using grass (HTC_GP_200 and HTC_GP_220) are consistent with the prominence of the second effect, since all N, NO$_2$ and NO$_3$ decrease at 220 °C. The high concentration of NO$_3$ for HTC_G_200 is outstanding. In the case of OS single HTC runs (HTC_OS_200 and HTC_OS_200), elemental analyses on the HC did not show any remarkable effect.

Including GP in the OS reaction system involves synergetic effects regarding the migration of metals to the solid phase at 200 °C, as it is the case for Mg, Al, P, and Ca, while Fe has a bigger concentration on the PW. The effect of co-HTC on NO$_3$ is substantial, involving a lower concentration if both biomass are used together than the one found for each one separately. Huang et al. also found a synergetic effect regarding mineral matter when co-hydrocarboning PVN wastes with pine sawdust [25].

With respect to the production of water, previous experimentation made by the authors on HTC of fresh grass with no water addition had previously shown that almost all the moisture of this biomass (nearly 80% *w/w*) was removed during HTC. From our results here, it was found that the amount of PW produced was dependent on the reaction conditions

and increased at higher temperature. Considering that 54 g of GP was used in each run, the amount of liquid obtained at 200 and 220 °C, was respectively 35.0 and 39.8 g; this corresponds to 68.8 and 73.7% of the biomass initial mass used, that is, the amount of water produced approaches the $H_2O$ amount of it as process temperature increases. Moreover, when OS is added (with 70 g of water) to GP, the net water production exceeds the found for single OS processes. A synergy is then found regarding the water use.

The balance about how much water is consumed and produced during HTC has scarcely been addressed in the literature. In a previous study, Reza et al. described that water-consuming processes (hydrolysis) and water-producing processes (dehydration) occur at different extents for a given feedstock depending on temperature, and the balance was found to evolve from negative to positive in the temperature range of 200–260 °C [26]. In the present study, this trend is negligible for single OS runs, but very clear for GP runs, and also for hybrid processes. When fresh GP is heated alone in the autoclave, the water vapor released as a result of drying (about 80% of its weight, that is, 40 g) will act as a reactant that at temperatures above 180 °C will start the hydrolysis of the glycosidic groups of hemicellulose and cellulose, yielding a wide range of products. After this, physical and chemical dehydration will contribute to an increase in processing water. Our results suggest an enhanced dehydration when GP is added to the system if a higher temperature is used; also, adding GP to OS slightly decreases the acidity of the processing water. Future works will be devoted to investigate if this change in processing water acidity caused by the addition of another feedstock has an effect on the process kinetics; the effect on SY seem to be slightly positive.

Analyses of the PW (results for 200 °C runs shown in Table 2) showed some synergetic effects of using the two biomasses together; for example, metals like Mg, Al, P, and Ca decreased their concentration for Co-HTC in relation to single processes (not finding this effect for Fe). N in liquid showed intermediate value, while the decrease of $NO_3$ for Co-HTC was also outstanding.

## 4. Conclusions

The HTC treatment of grass pruning residue at 200–220 °C (20 h) without a water addition can produce a liquid product with yield of about 80% and a solid carbon material with moderate HHV (22–25 MJ/kg). Under the same temperature conditions, olive stone was subjected to HTC, to yield a solid HC with HHV in the range 25–28 MJ/kg and a water production close to zero. Processing both biomasses in a co-HTC configuration demonstrated several advantages, including:

- At least 35% of water required for HTC of OS can be provided by GP. Neither the SY or the HHV are degraded by this modification.
- The net water production from HTC is improved when both materials are used together. Additionally, processing water from single OS HTC is more acidic than that from co-HTC.
- Slight changes on the N content on the HCs produced from both happen when they are used together. A slight increase in this element is found for olive stone HCs at expenses of the N of grass pruning.
- Co-HTC seems to enhance the precipitation of a wide amount on minerals on the HCs, decreasing the concentration of all the metals measured with the exception of Ca in the liquid phase.

Future research might investigate the change on the amount of water supplied by the high moisture biomass when the ratio between both precursors is changed, and the range at which it can guarantee suitable final properties of the dry feedstock. Separately, changes in time-dependent HTC processes as a result of the modifications of the properties of PW, because of the addition of the blending biomass to the system, should be evaluated.

The water saving attained by co-HTC is positive but may not be suitable if the toxicity of the resulting processing water is increased; the changes on water composition have to be

investigated. Moreover, the joint use of a biomass that can supply water to the process and the processing water recirculation will be addressed in future works.

**Author Contributions:** Conceptualization, S.R.; methodology, B.L.; investigation, R.G.-M.; resources, R.G.-M.; data curation, C.C.; writing—original draft preparation, S.R. and B.L.; writing—review and editing, R.G.-M.; supervision, C.C.; funding acquisition and editing, S.R. All authors have read and agreed to the published version of the manuscript.

**Funding:** This research was funded by to Agencia Española de Investigación for the financial help through project PID2020-116144RB-I00/AEI/10.13039/501100011033.

**Data Availability Statement:** Data sharing not applicable.

**Acknowledgments:** The authors are thankful the SAIUEX (Servicios de Apoyo a la Investigación de la UEX) for their support on the characterization analysis.

**Conflicts of Interest:** The authors declare no conflict of interest.

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
