# Peer review of "Co-Hydrothermal Carbonization of Grass and Olive Stone as a Means to Lower Water Input to HTC"

_resources, doi:10.3390/resources12070085_

Round 1

Reviewer 1 Report

The article is an interesting work on performing co-HTC. The article is well-written, and the methodology and discussions are complete – I suggest its acceptance after a minor revision. In general, authors should clarify when they are referring to dry or wet hydrochars so to avoid any misunderstanding to the reader. More specifically, the following points could help in improving the article.

Abstract

  • I suggest adding some numeric references in support of the comments on the results.

Methods

  • Lines 106-108. You could add details on the moisture content of the two biomasses to help the reader in following the speech. 
  • Lines 112-114. You should clarify if the second HTC reaction was performed using the dry hydrochar of GP or the entire slurry (dry hydrochar plus water). To avoid any possible misunderstanding, I suggest using a more clear nomenclature for the different terms.  
  • Figure 1. Is the water of GP “produced” or released/desorbed from the initial substrate (or both)?
  • Line 123. I suggest adding if SY is computed on a dry or wet based on the starting feedstock.

Results

  • Section 3.1. I think that the most interesting (and new) part of the section is the yield of the processed water. I suggest adding some considerations to the discussion so as to value the novelty of the paper.
  • Type note: the title “section 3.1.1.” is missing.
  • Lines 175-178. I think it would be useful to expand the point on why GP undergoes a very high weight reduction. Is this weight reduction due to HTC reactions and the macromolecular nature of the biomass or simply due to a denominator of the solid yield that contains the wet feedstock? 
  • Table 1. Is HHV on a dry or wet base? I could add this detail inside the caption. I also suggest reporting WP as mass yield instead of grams. Also, there is a mix of commas and dots and nonuniform decimals (some have 1, some 2) – you should uniform everything. 
  • Figure 5. I suggest removing the entire black SEM bar and leaving only the scale.  
  • Lines 281-282. I suggest changing the word “vulnerable” (even if poetic) to a more technical term regarding thermal stability.

Author Response

Dear reviewer,

thank you very much for your support and help in improving our manuscript; attached you can find our letter, in which we address the answers to your suggestions,

yours sincerely,

Silvia Román

Reviewer 2 Report

This manuscript describes an experimental study aimed at co-treating two different waste biomasses (olive stones and grass pruning) through hydrothermal carbonization (HTC), saving some process water. The manuscript is clear and the study targets an existing problem in the HTC technology, proposing an approach that seems effective and has not received coverage in the scientific literature.

At the same time, I think there are some overarching problems related to HTC process water that the manuscript should at least address.

1) In the framework of a continuous industrial process, recirculating the process water would probably ensure a much more significant water saving: why was this aspect not investigated?

2) Moreover, the authors do not mention how they expect the produced process water should be valorised/disposed of. More in-depth analysis may perhaps prove that the co-HTC creates an effluent that is more difficult to treat.

In addition to these previous points, there are some other queries that the authors should address: please find them below.

3) The Introduction should mention that HTC process water has often been deemed problematic.

4) 20 h is a very long residence time compared to other studies: why was it chosen?

5) HTC experiments with grass pruning were performed without any water added. Was the biomass only partially submerged during the HTC(/VTC?) tests? Could this explain some of the differences observed in the co-HTC tests, in which some water was added from the beginning?

6) In the co-HTC tests, the authors state that they separated the two hydrochars manually. Is an acceptable separation of the two materials realistically possible? How was it performed?

7) Table 1: The number of decimals and the decimal separator should be aligned. Standard deviations should be included (same for Table 2). Is an analysis of the two biomasses on a dry basis available?

8) Lines 202-220 are a repetition.

9) Fig. 3b is unclear: the legend is incomplete and the caption is too concise.

10) Fig. 4 is also missing the legend.

11) Table 2: Why was the process water analysed only for some operative conditions?

12) The solid yield is calculated considering the wet biomass weight. This should be explicitly stated and commented.

The manuscript contains several typos. Most of them could be easily identified by a word processing software.

Author Response

We enclose as a pdf file our answers to reviewer 2; thank you very much to help us improve our manuscript.

Reviewer 3 Report

Dear authors,

thank you for the interesting mansucript. I recommend it for publication with a single remark:

- Why there is no chapter 4?

nothing to add here

Author Response

We really thank reviewer 4 for his/her motivating revision. We have renumbered our sections, and the new version of the manuscript has included a chapter 4; we just got wrong in numbering the sections.

Reviewer 4 Report

The paper “Lowering Water Input by Co-Hydrothermal Carbonization: When One Biomass Can Partially Supply the Water to the Other” presents interesting data on the hydrothermal carbonization (HTC) of two substrates (garden pruning residuals, GP; and olive stone, OS) when reacted/carbonized one at a time or rather together. Several analyses were performed on the hydrochars, namely elemental analysis, thermogravimetric analysis, also coupled with mass spectrophotometry, SEM, and FTIR. Also the HTC process water was characterized in terms of metal and C and N concentrations. As a whole, the authors did a great job in terms of characterization and data analysis.

That said, I have some concerns on the data presentations, and I suggest the authors to pay more attention on the use of the language which is in some cases weak and in others simply inaccurate. I suggest also a more concise title, maybe something like “Co-Hydrothermal carbonization of grass and olive stone as a means to lower water input to HTC”.

About data presentations, below some aspects which should be clarified/better explained.

Line 123: <>. I usually think SY in term of dry feedstock (dry hydrochar/dry input feedstock) but here it seems different: hydrochar (dry?)/raw feedstock. The authors should specify this openly in the text to make it clearer.

Line 133: << The elemental composition (C, H, N and S) of the two precursors was determined using a CHNS analyzer (LECO CHNS-932); O content was estimated as the difference between the sum of the analyzed elements and ash to 100%.>>. Again, the authors should be more specific on the basis (as received, dry, dry ash free: ?). Usually, ultimate analysis data are presented on a dry basis, but this does not seem the case looking at Table 1: the sum the percentages of C, H, N, and O gives 100 as a result, and no data about ashes are reported. The value of the oxygen content of the Raw GP is 80.9: this value is very high, so it should be on as a received basis. And what about hydrochars? The authors should clarify about the ultimate analyses data, as the way they are reported is not clear. And what about ashes?

Figure 2: what reported above reflects on the values of H/C and O/C in the Van Krevelen diagram.

Figure 4. Both the two subfigures lack a legend, which in the first case must associate a molecule (H2O, CO and CO2) to the color in the histogram, in the second case it is not actually clear what the histograms refer to. Be more specific and clear.

In general, the discussion about the results and the data by the authors is interesting and well performed. As advice I would reduce the emphasis on the fact that water consumption is reduced by doing the co-HTC of the two analyzed substrates. Actually, this is the case, but I don't think it is the most important thing of the article - and from this consideration of mine also my proposal of modifying the title, as suggested above.

The English is not bad but it should be improved. Inaccuracies are presents (e.g. line 408: <>).

Author Response

Dear reviewer,

We are very grateful for the time and effort you have devoted to improve our work; thank you very much. In the attached letter, we describe the changes we have made following your suggestions. 

Reviewer 5 Report

I have several questions or comments:

1. Why did you have 95.6% moisture for OS unlike GP having 80% moisture?

2. Can I have approximate test along the element analysis as you presented?

3. I couldn't find conductivity test (line #194-195) on Table 1.

4. Cannot download your attachment file Tabel S1 thru the link you provided (#451-453).

5. Some mistakes on the table 1: comma --> dot, co-HC_GP_200 --> co-HC_GP_220.

6. How come pHmixture >pHGP or pHOS? 

7. I believe the Fig 2 is not corresponding with the data of the Table 1. If that is true the line #221-231 should be written again.

8. For the Fig 3 (b) and Fig 4, need more explanation.

9. On Table 2, can you explain why Mg, P, NO3 values for co_HTC_OC_200 is unreasonably low?

10. Hopefully I want more strong conclusions.

Pls check your spellings and sentensing. Below is few example I found.

1. the line 202-220 has been repeated.

2.  fressh --> fresh (#324), deammination -->deamination (#244), vIn -->In (#355), expklerimentation --> experimentation (#412)

Author Response

We thank reviewer 5 for his/her help to make our manuscript better; at the attached letter we have included, point by point, the explanations about how we have addressed the suggestions.

Round 2

Reviewer 5 Report

Pls check below for comments & questions:

1. In the Table 2, HC-OS_220 H = 4.0. Is this correct value?

2. Maybe I am misunderstanding but... co-HC_OS_200 should be same as co-HC_GP_200 because they are the same mixture of OS and GP? If not, what is the difference between two.  Same story for co_HC_220 and co-HC_GP_220. 

spelling check please.

1. #25 3-3% -->3.5%

2. #48, #49  (such --> such..... (Roman 

3. Table 1 co_HC-GP_220  N=3.4 O=31.7 HHV=25.9. These three values should be corrected.

Author Response

Dear reviewer,

thank you one more for your dedication to our work, that has definitely been very important to improve the final version of our paper. In the new version of the manuscript you can see that we have answered to your suggestions.

Regarding your question, we have blended the two biomass feedstock, but after the HTC processes we have separated them again, so that for each HTC process we have 3 products: two HCs (one from each starting material) and one type processing water (impossible to separate the fraction of each feedstock). That is why for each co-HTC sample there are two different HC properties (one for GP, another one for OS). 

Regarding the other issues, we have followed them. We are sorry we didn't´t realize the data about co-HC_220_GP had not been updated correctly; it has been done, in the new version of the manuscript.

We have also corrected the typos.

 Thank you very much again, yours sincerely,

Silvia Román